# Exploring Connections between Oral Microbiota, Short-Chain Fatty Acids, and Specific Cancer Types: A Study of Oral Cancer, Head and Neck Cancer, Pancreatic Cancer, and Gastric Cancer

**DOI:** 10.3390/cancers15112898

**Published:** 2023-05-24

**Authors:** Zahra Nouri, Sung Weon Choi, Il Ju Choi, Keun Won Ryu, Sang Myung Woo, Sang-Jae Park, Woo Jin Lee, Wonyoung Choi, Yuh-Seog Jung, Seung-Kwon Myung, Jong-Ho Lee, Joo-Yong Park, Zeba Praveen, Yun Jung Woo, Jin Hee Park, Mi Kyung Kim

**Affiliations:** 1Cancer Epidemiology Branch, Division of Cancer Epidemiology and Prevention, National Cancer Center, 323 Ilsandong-gu, Goyang-si 10408, Gyeonggi-do, Republic of Korea; zhnouri@ncc-gcsp.ac.kr (Z.N.); 2107102@ncc.re.kr (Z.P.); jhblue99@ncc.re.kr (J.H.P.); 2Oral Oncology Clinic, Research Institute and Hospital, National Cancer Center, 323 Ilsandong-gu, Goyang-si 10408, Gyeonggi-do, Republic of Korea; choiomfs@ncc.re.kr (S.W.C.); leejongho@ncc.re.kr (J.-H.L.); slowp@ncc.re.kr (J.-Y.P.); 3Center for Gastric Cancer, National Cancer Center, 323 Ilsandong-gu, Goyang-si 10408, Gyeonggi-do, Republic of Korea; cij1224@ncc.re.kr (I.J.C.); docryu@ncc.re.kr (K.W.R.); 4Center for Liver and Pancreatobiliary Cancer, National Cancer Center, 323 Ilsandong-gu, Goyang-si 10408, Gyeonggi-do, Republic of Koreaspark@ncc.re.kr (S.-J.P.); lwj@ncc.re.kr (W.J.L.); 5Center for Rare Cancers, National Cancer Center, 323 Ilsandong-gu, Goyang-si 10408, Gyeonggi-do, Republic of Korea; wonyoungchoi@ncc.re.kr; 6Department of Otorhinolaryngology, National Cancer Center, 323 Ilsandong-gu, Goyang-si 10408, Gyeonggi-do, Republic of Korea; jysorl@ncc.re.kr; 7Department of Cancer AI & Digital Health, National Cancer Center Graduate School of Cancer Science and Policy, 323 Ilsandong-gu, Goyang-si 10408, Gyeonggi-do, Republic of Korea; msk@ncc.re.kr

**Keywords:** oral microbiota, machine learning, SCFAs, FFAR2, TNFAIP8, IL-6/STAT3, cancer onset

## Abstract

**Simple Summary:**

Although there is strong evidence linking oral microbiota to several types of cancer, the causal connections between them remain controversial. This study aims to identify the common oral bacteria associated with various types of cancer and detect potential mechanisms underlying the oral microbiota that could activate immune responses and lead to the onset of cancer through cytokine secretion. We have confirmed that alterations in the composition of oral bacteria can contribute to a reduction in SCFAs and the expression of the FFAR 2, resulting in an inflammatory response through the upregulation of TNFAIP8 and the IL-6/STAT3 pathway, ultimately increasing the risk of cancer onset. These findings provide valuable insights into the potential role of the oral microbiome in cancer development and could pave the way for novel preventive and therapeutic strategies for cancer.

**Abstract:**

The association between oral microbiota and cancer development has been a topic of intense research in recent years, with compelling evidence suggesting that the oral microbiome may play a significant role in cancer initiation and progression. However, the causal connections between the two remain a subject of debate, and the underlying mechanisms are not fully understood. In this case–control study, we aimed to identify common oral microbiota associated with several cancer types and investigate the potential mechanisms that may trigger immune responses and initiate cancer upon cytokine secretion. Saliva and blood samples were collected from 309 adult cancer patients and 745 healthy controls to analyze the oral microbiome and the mechanisms involved in cancer initiation. Machine learning techniques revealed that six bacterial genera were associated with cancer. The abundance of *Leuconostoc*, *Streptococcus*, *Abiotrophia*, and *Prevotella* was reduced in the cancer group, while abundance of *Haemophilus* and *Neisseria* enhanced. G protein-coupled receptor kinase, H+-transporting ATPase, and futalosine hydrolase were found significantly enriched in the cancer group. Total short-chain fatty acid (SCFAs) concentrations and free fatty acid receptor 2 (FFAR2) expression levels were greater in the control group when compared with the cancer group, while serum tumor necrosis factor alpha induced protein 8 (TNFAIP8), interleukin-6 (IL6), and signal transducer and activator of transcription 3 (STAT3) levels were higher in the cancer group when compared with the control group. These results suggested that the alterations in the composition of oral microbiota can contribute to a reduction in SCFAs and FFAR2 expression that may initiate an inflammatory response through the upregulation of TNFAIP8 and the IL-6/STAT3 pathway, which could ultimately increase the risk of cancer onset.

## 1. Introduction

Cancer is one of the leading causes of death worldwide, and despite significant progress in treatment and prevention, early detection remains a critical challenge. For many years, epidemiological studies have reinforced several cancer risk factors, including heredity, diet, age, and inflammation [1]. Moreover, studies have also examined how certain lifestyle factors, such as smoking, alcohol consumption, and a high body mass index (BMI), may increase the risk of developing upper-digestive tract, gastric, liver, pancreatic, and breast malignancies [2,3]. Early cancer detection reduces medical expenses as expensive and long-term treatments accompanying later disease detection are generally not required [4,5]. A conventional cancer diagnosis still depends on the prompt reporting of symptoms and single-tissue medical imaging, followed by a histopathological analysis of tumor biopsies. Such processes are generally ineffective for early cancer diagnoses as they rely on symptomatic and phenotypic changes typically appearing at later malignancy stages [4]. However, new sensitive and precise detection methods for multiple cancer types have emerged that can improve early cancer diagnosis rates, and ultimately benefit population health and economics [5].

The oral microbiota consists of complicated ecosystems which have pivotal roles in maintaining homeostasis, adjusting immune responses, and resisting disease in the body [6]. Previous human research studies reported a possible link between the oral microbiota and carcinogenesis in different organs, including the gastrointestinal tract, the head and neck, the oral cavity, and the pancreas [7,8,9,10,11]. In line with these findings, a positive association of a higher diversity of microbiota with *Haemophilus*, *Porphyromonas*, *Leptotrichia*, and *Fusobacteria* compared to healthy adults with pancreatic cancer has been observed in large cohort human studies [12,13]. Additionally, investigations into *Lactobacillus* and *Streptococcus* genera (oral bacteria) have shown that they generate volatile sulfur compounds, short-chain fatty acids (SCFAs), reactive oxygen species, reactive nitrogen species, hydrogen peroxide, and lactic acid, all of which are implicated in carcinogenesis, chronic inflammation, genomic instability, tumor angiogenesis, and GC progression [14]. Recent studies have also reported that *Porphyromonas gingivalis* promotes OC development and progression by stimulating oral squamous cell carcinoma proliferation and inducing the expression of key molecules nuclear factor kappa B (NF-κB), interkeukin-6 (IL-6) signal transducer and activator of transcription 3 (STAT3), cyclin D1, matrix metallopeptidase 9 (MMP-9), and the bacterial gingipains which are implicated in tumorigenesis. In gut microbiota-related cancer, previous data have suggested that the free fatty acid receptor 2 (FFAR2) and SCFAs (propionate, butyrate, and acetate), as the most significant microbiota metabolites, act on G protein-coupled and are involved in pro-inflammatory cytokine generation, intestinal immune reactions, and reducing carcinogenesis [15,16]. It was reported that obesity constituted a link between dysbiotic *Hafnia alvei* and *Akkermansia muciniphila* bacteria in the gastrointestinal tract and lower SCFA abundance, particularly butyrate. These effects activated FFAR2 and enhanced cytokine expression, especially tumor necrosis factor-α (TNF-α) and IL-6 which eventually induced gastrointestinal cancer [17,18,19]. To date, no studies have specifically examined associations between oral microbiota composition and specific cancer types, such as oral cancer (OC), head and neck cancer (HNC), pancreatic cancer (PC), and gastric cancer (GC). Additionally, no studies have yet investigated the potential role of oral microbiota composition in the initiation of multiple types of cancer. Despite the growing body of research on the association between oral microbiota and cancer, several gaps in our understanding still exist. Some studies have reported conflicting findings regarding the specific bacteria that are associated with cancer initiation, while others have found variations in the bacterial profiles of cancer patients depending on the type and stage of cancer. Additionally, previous studies have been limited in scope, often focusing on a single type of cancer or a specific set of oral microbiota. Furthermore, the exact mechanisms by which oral microbiota contribute to cancer initiation and progression are still not fully understood. While some studies have suggested that certain bacteria can directly induce oncogenic changes in host cells, others have proposed that the host’s immune response to bacterial colonization and inflammation plays a key role in cancer onset. We hypothesized that crosstalk between the oral microbiota and mechanisms underlying immune factors associated with SCFA alterations in the initiation of specific cancer types, such as OC, HNC, PC, and GC. To address this, we identified the common oral microbiota and explored the correlation between oral bacteria and possible orthologs found in the Kyoto Encyclopedia of Genes and Genomes (KEGG) pathways associated with several types of cancer. Subsequently, we explored the potential mechanisms of the oral microbiota that can prompt immune responses and initiate cancer upon cytokine secretion.

## 2. Materials and Methods

### 2.1. Subject Characteristics

In this case–control study, we enrolled adult cancer patients (aged > 19 years) with newly diagnosed histologically confirmed invasive cancers, who had not received any therapy, surgery, or used immunosuppressive treatment agents. Participants came from the National Cancer Centre, and Seoul National University Dental Hospital, Republic of Korea. Healthy controls with no cancer at enrolment were recruited from the Cancer Screening Centre, National Cancer Centre, Republic of Korea. Eligible subjects were identified from 309 cancer patients with oral cancer (n = 178), HNC (n = 21), PC (n = 50), and GC (n = 60), and 745 healthy controls.

Among characteristic parameters, age, sex, smoking and drinking status, stage, tumor extent (T stage), and lymph node involvement (N stage) were considered as the main clinical variables. BMI was constituted of continuous variables and divided into 5 groups: lean with less than 18.5, normal with 18.5 to 22.9, overweight with 23.0 to 24.9, obese with 25.0 to 29.9, and severely obese with 30 or more value of BMI. The smoking status of participants was classified into three groups: non-smoker, current smoker, and former smoker, and in terms of drinking, they were divided into three groups: non-drinker, current drinker, and former drinker. All cancer was classified into four stages. Regarding cancer stages, T stage was classified as (T1, T2, T3, and T4), while N stage was categorized into (N0, N1, N2, and N3). The study was approved by the Institutional Ethics Committee of the National Cancer Centre, Korea (IRB No. NCC 2019-0116), and the Seoul National University Dental Hospital (IRB No, CRI15017). Participants provided written informed consent prior to enrolment.

### 2.2. Saliva and Blood Sampling

From our comprehensive clinical protocol, saliva, and blood samples were collected. For the saliva samples, participants were asked to refrain from eating, drinking, or smoking for 1 h before collection. Saliva was then deposited into a specimen cup, aliquot into 1.5 mL tubes, and stored at −80 °C. Blood samples were collected after a 12 h fast from the antecubital veins using K2 EDTA tubes (BD Vacutainer, Franklin Lakes, NJ, USA). Tubes were centrifuged at 3000 rpm for 15 min at 4 °C to generate, plasma, buffy coat, and red blood cells were stored at −80 °C. These measures allowed us to analyze the microbiota and other factors in the saliva and blood samples between cohorts at baseline.

### 2.3. DNA Isolation, 16S rRNA Gene Sequencing, and Analysis

Oral microbial DNA was extracted from saliva samples using the Fast DNA Spin extraction kit (MP Biomedical, Santa Ana, CA, USA) according to manufacturer’s instructions. DNA quality and quantification were evaluated using a Qubit dsDNA BR assay kit (Life Technologies, Carlsbad, CA, USA) on a Qubit fluorometer (Life Technologies). Polymerase chain reaction (PCR) thermal cycle parameters were 95 °C for 3 min, and 25 cycles of 95 °C for 30 s, 55 °C for 30 s, and 72 °C for 30 s, with a final elongation at 72 °C for 5 min. Following 2% agarose gel electrophoresis, PCR products were visualized and purified products were used for secondary amplification steps to attach the Illumina (San Diego, CA, USA) NexTera barcodes using Index 2 i5 forward and Index 1 i7 reverse primers (Bionics, Cosmogenetech, Seoul, Republic of Korea) (Appendix A). Reactions underwent eight cycles in aforementioned thermal cycling conditions, followed by PCR product purification using an AMPure bead kit (Agencourt Bioscience, Beverly, MA, USA). Mixed amplicons were pooled in Chunlab (https://www.cjbioscience.com/ (accessed on 21 August 2022)) and DNA isolation and sequencing were performed at the National Cancer Center South Korea using an Illumina iSeq100 sequencer. The variable V4 region of the bacterial 16S rRNA gene was amplified using barcoded fusion primers 341F and 805R (Bionics, Cosmogenetech, Seoul, Republic of Korea). The Ezbiocloud cloud database holds taxonomic information and functions as a bioinformatic tools to help classify bacteria. Poor quality sequences of reading length < 80 bp or >2000 bp were eliminated and averaged Q values were <25. DUDE-Seq. software was used for denoising and the identification of non-redundant reads. The UCHIME algorithm was used against the Ezbiocloud 1616s-based microbial taxonomic profiling database to evaluate and eliminate chimeric sequences. Taxonomic assignments were performed using USEARCH tools to find and quantify sequence homology in query single-end reads against the EzBioCloud 16s-based microbial taxonomic profiling database. Sequencing reads were classified into operational taxonomic units (OTUs) with 97% sequence similarity using the UPARSE algorithm. To cluster single-end read samples into multiple OTUs, the UCLUST tool, with aforementioned cutoff values, was used.

### 2.4. Measuring SCFA, FFAR2, and Chemokine/Cytokine-Related Cancer Associations with Oral Microbial Signals

Total SCFAs in saliva were measured using a human SCFA enzyme-linked immunosorbent assay (ELISA) kit (Catalog No. MBS7269061; MyBioSource, Inc., San Diego, CA, USA). According to manufacturer’s instructions, different standard concentrations and saliva samples (100 μL) were added to 96-well plates. Next, 10 μL balance solution was added to only the samples. Then, 50 µL conjugate was added to wells (not the blank control well). The plate was covered and incubated for 1 h at 37 °C. A second incubation step occurred for 15–20 min at 37 °C following the addition of 100 μL substrate A and B. Finally, 50 μL stop solution was added to wells and optical density was determined using a microplate reader (SPECTROstar Nano, Ortenberg, Germany) at 450 nm.

Human plasma FFAR2 (Catalog No. abx556340; Abbexa Ltd., Cambridge, UK) and IL-6 (Catalog No. ab178013; Abcam, Cambridge, UK) levels were determined using ELISA kits according to manufacturer’s instructions. The minimum detectable concentration was 8 pg/mL for FFAR2 and 1.6 pg/mL for IL-6. Plasma STAT3 concentrations were also determined using ELISA kits (Catalog No. MBS2509698; Mybiosource, San Diego, CA, USA); its lower limit of detection was 0.19 ng/mL. Plasma TNFAIP8 levels were also quantified using ELISA (Catalog No. ABIN6951264; Antibodies-Online, Limerick, PA, USA); its minimum detectable concentration was 0.188 ng/mL.

### 2.5. Oral Microbiota Key Metabolic Pathways Prediction

Oral microbiota functional profiles were evaluated using the phylogenetic investigation of communities by reconstruction of unobserved (PICRUST) algorithm in the EzBioCloud 16S-based microbiota taxonomic profiling pipeline. Raw sequencing reads were obtained by running the EzBioCloud 16S microbiota pipeline with default parameters and discriminating reads within the reference database. The annotation of oral microbiota functional abundance profiles was based on bioinformatics, specifically by multiplying the vector of gene counts for abundance of that OTU value of each taxon, using the KEGG orthology module and pathway database. Based on microbial anticipation in results, discrepancies in KEGG pathways, and KEGG orthologies related to metabolism and biological systems between the different cancer groups were detected. The accuracy of each functional profile was determined using the nearest sequenced taxon index.

### 2.6. Machine Learning and Statistical Analyses

Machine learning was performed using Python version 3.7.15 (Python Software Foundation) and the H2O python module (Python package version 3.38.0.2. https://github.com/h2oai/h2o-3 (accessed on 3 January 2023)). To address class imbalance, oversampling methods were applied using the “smote-variants” package (version 20), which included “Borderline-SMOTE”, “Safe-level-SMOTE”, “polynom-fit-SMOTE”, “ADASYN”, and “SYMPROD” [20,21,22]. These methods generate synthetic samples for the minority class to balance the dataset. Specifically, Borderline-SMOTE generates synthetic samples near the decision boundary, Safe-level-SMOTE focuses on safe-level minority samples, polynom-fit-SMOTE fits a polynomial function to the dataset to generate synthetic samples, ADASYN adjusts the density distribution of the minority class to generate synthetic samples, and SYMPROD applies synthetic minority over-sampling in a product space. To exclude low-count taxa, only groups with at least 10 genera and significance as indicated in the Wilcoxon rank sum tests were included. Three algorithms were used: generalized linear model (GLM), random forest (RF), and gradient-boosting machine (GBM), and their hyper parameters were fine-tuned. For the GLM, hyper parameter tuning was performed by systematically varying the regularization strength and type to optimize model performance. Regularization is a technique that helps prevent overfitting by adding a penalty term to the cost function that shrinks the magnitude of the coefficients. Three types of regularization were used: LASSO (L1 penalty), ridge (L2 penalty), and elastic net (a combination of both penalties). The hyper parameters were optimized using cross-validation, which involved splitting the data into training and validation sets and evaluating model performance on the validation set. The hyper parameters that resulted in the highest F2 scores and AUC-PR were selected. For the RF and GBM, hyper parameter tuning was also performed to optimize the performance of the models. Specifically, the complexity of individual trees and their structures were varied to find the best-performing models. The performance of the classifiers was evaluated using F2 scores, the area under the precision-recall curve (AUC-PR), AUC, and accuracy metrics, with a focus on the positive class.

Analyses and visualizations were performed in R version 4.1.1 (R Foundation for Statistical Computing, Vienna, Austria). General patient characteristics were compared between groups using *t*-tests for continuous variables (age and BMI) and chi-square tests for categorical variables (BMI, smoking status, drinking status, stage, T stage, and N stage). Groups were analyzed using one-way analysis of variance for continuous variables and chi square tests. Alpha diversity indicated bacterial richness and diversity using OTU and phylogenetic diversity (PD) whole tree analyses. Beta diversity was calculated using principal coordinates analysis (PCoA) according to weighted and unweighted UniFrac distances, and evenly sampled OTU abundance. Results were analyzed using the “Phyloseq” package in R. Statistical significance in groups, taxa, and functional composition profiles was analyzed based on abundance quartiles values, Wilcoxon rank sum tests, and fold change (FC) in R. Univariate logistic and multivariate-adjusted conditional logistic regression analyses were conducted in R. Logistic regression analyses were conducted to estimate odds ratios (OR) and corresponding 95% confidence intervals (CIs). Adjusted conditional logistic regression was set for sex, age, smoking, and alcohol consumption. Differential abundance and cladogram analyses were performed using the linear discriminant analysis of effect size (LEfSe) method for taxa at the genus level in the “microbiota Marker” package in R. Venn diagrams identified overlaps between groups at the genus level using “Venn Detail” in R. The distribution of the six most important genera according to group, sex, drinking, and smoking status was performed using “ggplot2” and “ggpubr” in R. SCFA level, FFAR2 expression, and inflammatory cytokine correlations between distinctive oral microbiota, smoking, and drinking were performed using Wilcoxon rank sum tests in R. Associations between genera and orthology were analyzed using Spearman’s correlations in R (Figure 1).

## 3. Results

### 3.1. Demographic and Lifestyle Characteristics of Study Participants

In total, 1054 participants were selected, including 178 individuals with oral cancer, 21 with HNC, 50 with PC, 60 with GC, and 745 healthy controls. Participants were chosen based on inclusion and exclusion criteria to examine the changes in oral microbiota across different groups. The age and sex of participants in the case and control groups were matched according to the criteria. There were some minor differences in the age and sex distribution and oral dysbiosis between the case and control groups, and the cases had significantly higher rates of drinking, smoking, and overweight compared to the control group. (Table 1). The demographics and clinical characteristics of the four cancer groups and healthy controls and association between oral cancer risk and six genera with age and sex are summarized (Appendix A).

### 3.2. Taxonomic Analyses and the Identification of Genera Potentially Associated with Cancer Risk

In taxonomic analyses of the oral microbiota using machine learning, among six genera, GBM was the best model for all cancer prediction (F2 = 0.96, AUCPR = 0.98, sensitivity = 0.96, precision = 0.95, AUC = 0.98, and accuracy = 0.97). The disease index probability was significantly higher in the cancer group when compared with controls (Figure 2a–f). A greater abundance of *Leuconostoc* (FC = not available (NA); *p* < 0.01), *Streptococcus* (FC = 1.75; *p* < 0.01), and Abiotrophia (FC = 3; *p* < 0.01) genera was associated with a significant increase in the risk of all cancers, while *Prevotella* (FC = 0.67; *p* < 0.01), *Haemophilus* (FC = 0.56; *p* < 0.01), and *Neisseria* (FC = 0.76; *p* < 0.01) abundance was associated with decreased risk for all cancers (Table 2 and Appendix A).

### 3.3. Oral Microbiota Profiles Are Switched in All Cancer and Control Groups

In total, 50 phyla and 1666 genera were identified across all samples. From phylum level analyses (mean relative abundance > 1%), *Firmicutes*, *Proteobacteria*, *Bacteroidetes*, *Fusobacteria* and, *Actinobacteria* were the five most abundant phyla and accounted for > 90% of the bacterial community (Appendix A). The average number of readings per sample was 39,000, with a minimum of 25,000 readings per sample. The taxonomic analysis revealed that the percentage of readings assigned to each taxonomic group were as follows: *Protoebacteria* (50%), *Firmicutes* (15%), *Bacteroidetes* (12%), *Actinobacteria* (10%), and other groups (13%). We found no significant variation in the percentage of unassigned readings between the study groups. To examine oral microbiota fluctuations in cancer patients when compared with controls, we conducted LEfSe on 209 genera. To exclude low count taxa, we only included genera with at least one sequence in at least 5% of the participants. In alpha diversity analyses, PD whole tree and observed OTUs were significantly higher in all cancer cases when compared with controls (Wilcoxon, *p* < 0.001) (Figure 3a). Moreover, beta diversity, as shown by PCoA based on weighted and unweighted (Wilcoxon, *p* < 0.001) UniFrac distances in oral microbiota communities, exhibited clear and distinct divisions between groups (Figure 3b). An OTU Venn diagram showed that 722 genera overlapped between groups (Figure 3c). LEfSe cladogram results identified thirteen separated oral bacteria in the cancer groups and seven in the control group (Appendix A). At the genus level, *Streptococcus*, *Abiotrophia*, and *Leuconostoc* were significantly associated with an increased risk of all cancer, with higher risks identified in the highest quartile when compared with the lowest ones. In contrast, *Prevotella*, *Haemophilus*, and *Neisseria* showed a significant negative association with higher risks in the lowest quartile when compared with the highest. We further investigated oral microbiota associations in smoking and drinking subgroups. Correlations between distinctive bacteria, smoking, and drinking status are explained in Appendix A.

### 3.4. Identification of Potential Correlation between Specific KOs in Oral Microbiota and Increased Cancer Risk

In this study, a total of 14,842 KEGG KOs were identified by conducting PICRUSt analysis. Among this, G protein-coupled receptor kinase, H+-transporting ATPase, and futalosine hydrolase were found significantly enriched in the cancer group (Table 3). In the metabolic pathway of H+-transporting ATPase and futalosine hydrolase, SCFAs and its associated receptor (FFAR2) were found significantly down-regulated in the comparison between cancer and control group (*p* < 0.001) (Figure 3d,e) Subsequently, we explored the associations between six genera and three KOs based on Spearman’s rank correlation coefficients. The G protein-coupled receptor kinase (K08291), H+-transporting ATPase (K01535), and futalosine hydrolase (K1178), were remarkably correlated with oral microbiota and associated with cancer (Table 4).

### 3.5. Oral Microbiota Modified Systemic Inflammation and Cancer Initiation

To confirm the key role of the identified orthologs (K08291, K01535, and K11783) that correlated with oral microbiota in the cancer group and determine the effect of oral microbiota alteration on immune homeostasis, inflammation, cancer initiation, we measured the concentrations of TNF-α, induced protein 8 (TNFAIP8/TIPE), and IL-6, which are cytokines associated with these orthologies. The cancer group had higher TNFAIP8 serum (Wilcoxon, *p* < 0.001) (Figure 3f) and IL-6 levels (Wilcoxon, *p* < 0.001) (Figure 3g) when compared with the control group. We also measured STAT3 levels which were induced by IL-6. Similar to inflammatory cytokines, STAT3 had higher levels in the cancer group when compared with the control group (Wilcoxon, *p* < 0.001) (Figure 3h).

## 4. Discussion

In this first case–control study of the oral microbiota and risk for all cancers, we found that a greater abundance of the bacterial genera *Streptococcus*, *Abiotrophia*, and *Leuconostoc* while a lower abundance of *Prevotella*, *Haemophilus*, and *Neisseria* was associated with an increased risk of cancer. Furthermore, our analysis revealed that certain KOs, found to be significantly associated with oral microbiota, may contribute to the initiation of cancer by activating the SCFA and inflammation cytokine pathways. These findings highlight the important role of bacterial metabolites, such as SCFAs, in cancer onset and suggest potential targets for intervention in cancer prevention and treatment.

Even though ours is the first study examining the associations between the oral microbiota and all cancer risk, evidence in the literature supports this finding. Several culture and DNA sequencing-based studies identified oral microbiota prevalence in multiple cancer types while accounting for smoking and alcohol consumption influences [23]. In our novel cohort study, *Streptococcus*, *Abiotrophia*, and *Leuconostoc* were enriched while *Haemophilus*, *Neisseria*, and *Prevotella* were reduced in cancer patients. Alterations in oral microbiota communities, particularly *Firmicutes* and *Bacteroidetes* in alcohol consumers [24,25,26] and smokers, were associated with a risk of several cancer types [27,28]. For oral cancer, the evidence now shows that the well-known risk factors of smoking and alcohol consumption cannot explain 15% of oral cancer cases [29], which may be related to ecological shifts in microbiota abundance induced by homeostatic microflora loss [30,31]. These data suggest that smoking and drinking may reshape the oral microbiota, including the potential depletion of beneficial commensal bacteria and the increased colonization of potential pathogens linked to enhanced systemic inflammation and carcinogenesis [27].

Based on our results, oral microbiota and microbial compositional diversity were increased in all cancer groups concomitant with unhealthy lifestyles. This was concluded based on the enhancement of the relative abundance of some bacteria, including *Streptococcus*, *Abiotrophia*, and *Leuconostoc*, and the reduction in the relative abundance of *Prevotella*, *Haemophilus*, and *Neisseria* in the cancer group when compared with controls. Similar to other studies, patients with oral squamous cell carcinoma, hematologic malignancy, and esophageal cancer [32,33,34] had significant changes in saliva microbial diversity, such as *Streptococcus* [35,36,37] and *Abiotrophia* [38], when compared with healthy individuals. In contrast, *Leuconostoc*, believed to have probiotic properties and anticancer activities [39], promoted apoptosis in colon cancer cell lines by upregulating mitogen-activated protein kinase 1 (MAPK1), Bax, and caspase 3, and by downregulating AKT serine/threonine kinase 1(AKT), NF-κB, and B-cell lymphoma-extra-large (Bcl-XL) expression [40]. Interestingly, distinct salivary microbiota composition was observed in patients with GC and colorectal adenocarcinoma, and exemplified by enriched putative pro-inflammatory taxa including *Streptococcus* [41,42], and significantly decreased oral microbiota, including *Haemophilus*, *Neisseria*, and *Prevotella*, which reduced nitrites and potentially induced the accumulation of carcinogenic compounds [43]. Indeed, different bacteria have been suggested as potential risk factors in several cancers [38,44]. We showed that fluctuations in certain oral bacteria genera, such as *Streptococcus*, *Abiotrophia*, *Leuconostoc*, *Haemophilus*, *Neisseria*, and *Prevotella*, can produce SCFAs, which have been shown to induce cytokine production and inflammation. The activation of cytokine receptors through chronic inflammation and cytokine production can promote carcinogenesis. It has been highlighted that certain bacteria, such as *Capnocytophaga gingivalis*, *Prevotella melaninogenica*, and *Streptococcus mitis*, contribute to cancer initiation through various mechanisms, including the induction of chronic inflammation, interference with cellular signaling pathways, or metabolizing carcinogenic substances [29].

We found via in-depth investigations that among 14,842 KEGG KOs, 3 orthologs genes including, H+-transporting ATPase, G protein-coupled receptor kinase and, futalosine hydrolase were highly correlated with *Streptococcus*, *Abiotrophia*, *Leuconostoc*, *Haemophilus*, *Neisseria*, and *Prevotella*. The H+-transporting ATPase and futalosine hydrolase orthologs are involved in various aspects of cellular metabolism, particularly transport of SCFAs by activating FFAR2 across the cell membrane (https://www.genome.jp/pathway/map01100+K01535 (accessed on 14 October 2022)) (https://www.genome.jp/pathway/map01100+K11783 (accessed on 14 October 2022)). In the same context, this study suggested that alteration in specific bacteria may shift the expression of FFAR2 and SCFA, which could influence cancer initiation via cytokine receptor activation and chronic inflammation. Studies demonstrated that the fatty acid produced by the fermentation of *Faecalibacterium*, *Eubacterium*, and *Roseburia* is able to stimulate an immune response in mice in a GPR43-dependent manner. Moreover, activation of FFAR2 and SCFAs signaling pathway can stimulate a range of cellular responses, including the regulation of immune response and carcinogenesis [45]. The acetate loss induces lipid peroxidation following oxidative stress, resulting in gastric epithelial cell apoptosis and GC 54. Additionally, sodium acetate (>12.5 mM) impeded cell proliferation in PC cell lines (Capan-2, AsPC-1, and MiaPaCa-2) and caused cell detachment and reduced cell density [46]. Additionally, we showed that TNFAIP8, the IL-6 cytokine receptor, and IL-6-activated STAT3 pathway activity increased following FFAR2 reduction in the all cancer group. Several studies reported that elevated IL-6 and TNFAIP8 levels and reduced FFAR2 levels [19], and enhanced IL-6/JAK/STAT3 [47,48] and NF-κB [49] pathways. These pathways are aberrantly hyper-activated in many cancer types, the hyper-activation generally associated with poor clinical prognoses and cell proliferation. All findings indicate that alterations in the composition of oral microbiota lead to a reduction in levels of SCFAs and FFAR2. This reduction may contribute to the promotion of TNFAIP8 and IL-6/STAT3, ultimately provoking inflammation and potentially increasing the risk of developing cancer (Figure 4).

## 5. Conclusions

In conclusion, this study provided the first proof that alcohol consumption and smoking modified the oral microbiota, which contributed to cancer initiation and systemic inflammation. Smoking and drinking reduced the relative abundance of *Prevotella*, *Haemophilus*, and *Neisseria*, but increased *Streptococcus*, *Abiotrophia*, and *Leuconostoc*. These transformations in the oral microbiota can contribute to changes in microbial metabolites particularly SCFAs, cytokines, and chemokines, which may trigger an inflammatory response and potentially increase the risk of cancer onset. Therefore, our microbiota study provided links between microbiota composition, drinking, smoking, and carcinogenesis. By investigating the potential mechanisms behind these associations, our study sheds light on the complex interplay between oral microbial communities, immune response, and cancer development in several cancer types. However, further studies are needed to fully elucidate the functional connection between the oral microbiota and cancer development, and to determine whether the changes observed in this study are host-derived, dysbiosis-derived, or a combination of both. This information can be used to develop new and more effective prevention and treatment strategies. Early detection and monitoring of these microbiota changes could potentially allow for early intervention and targeted treatments, which may improve patient outcomes and reduce the burden of cancer on healthcare systems.

## Figures and Tables

**Figure 1 cancers-15-02898-f001:**
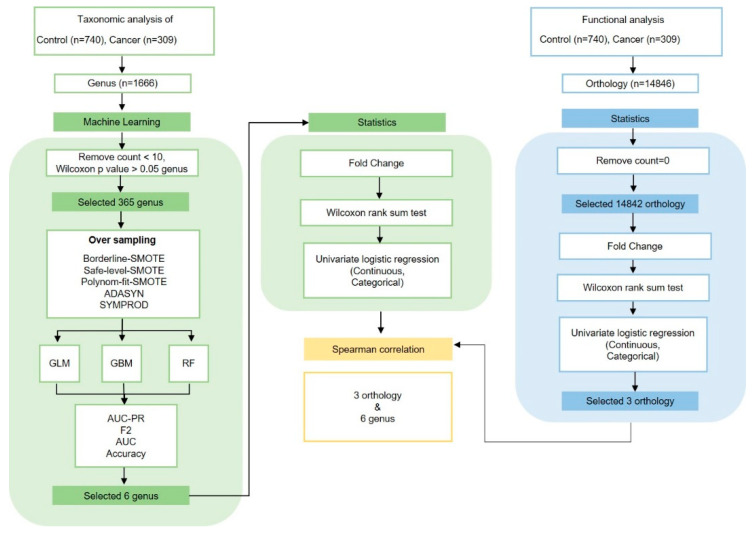
The study utilized advanced techniques, such as PICRUST and machine learning, to analyze oral microbiota, identifying correlations between certain microbiota and cancer, and suggesting potential early markers of cancer.

**Figure 2 cancers-15-02898-f002:**
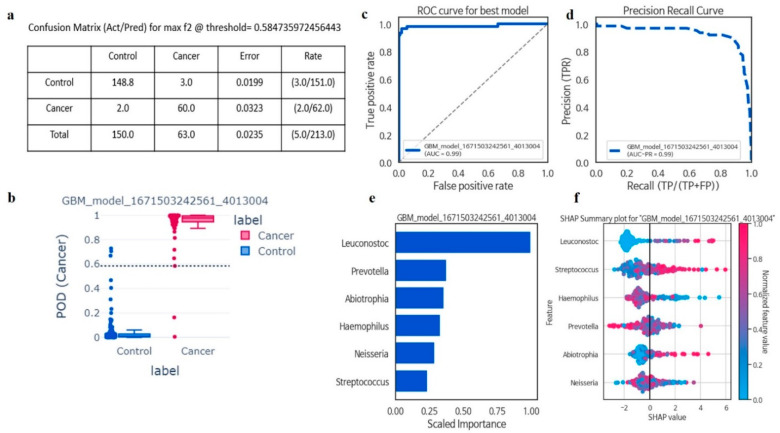
The six most important microbiota in the Gradient Boosting Machine (GBM) model, along with relevant classification information. Confusion matrix evaluated machine learning model performance with actual and predicted labels, and error rate calculated by incorrect predictions divided by total predictions (**a**); The Probability of Detection Index (POD) for Gradient Boosting Machine (GBM) displayed value (0–1) for classifying samples into control and cancer, with values closer to 1 indicating higher likelihood of being classified as cancer (**b**); The stands for Area Under the Receiver Operating Characteristic curve (AUC-ROC) represented the relationship between the true positive rate (TPR) against the false positive rate (FPR) at various classification thresholds (**c**); Precision–Recall Curve depicted binary classification model performance, plotting precision against recall at different classification thresholds, useful for imbalanced datasets (**d**); Feature importance indicated how much each feature contributes to model prediction, indicating the relative importance of features in distinguishing between classes (**e**); SHAP value revealed an explanation of machine learning model output, providing local interpretations for individual predictions by attributing the feature contribution to the final prediction (**f**).

**Figure 3 cancers-15-02898-f003:**
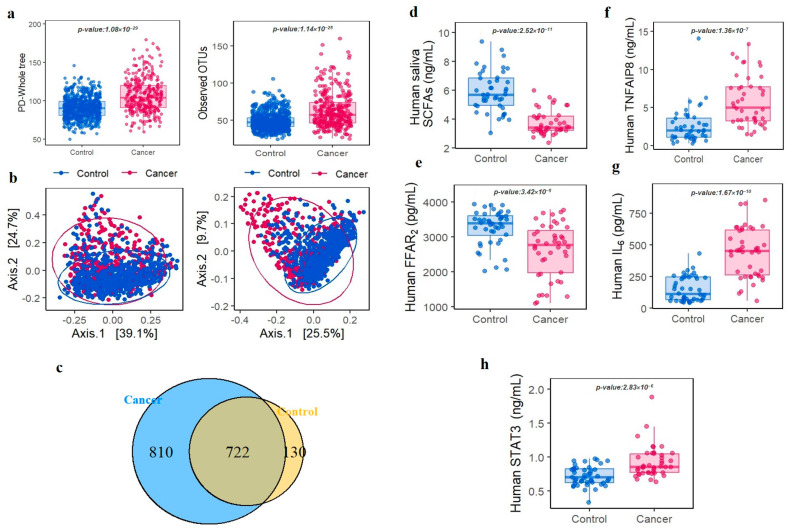
Analysis of oral microbiota diversity, SCFAs, FFAR2, and cytokine levels in the control and cancer groups. Alpha diversity was estimated via the phylogenetic diversity whole tree (OTUs, *p* < 0.001) and by observing operational taxonomic units (*p* < 0.001) (**a**); Beta diversity was calculated using principal coordinate analyses based on weighted (*p* < 0.001) and unweighted (*p* < 0.001) UniFrac distances in oral microbiota communities (**b**); Venn diagrams showing overlaps between groups at the genus level (**c**); Oral saliva total short-chain fatty acid concentrations (**d**); Free fatty acid receptor 2 concentrations in oral saliva (**e**); Concentrations of human plasma TNF-α induced protein 8 (**f**); Human plasma interleukin-6 levels (**g**); Human plasma signal transducer and activator of transcription 3 levels (**h**).

**Figure 4 cancers-15-02898-f004:**
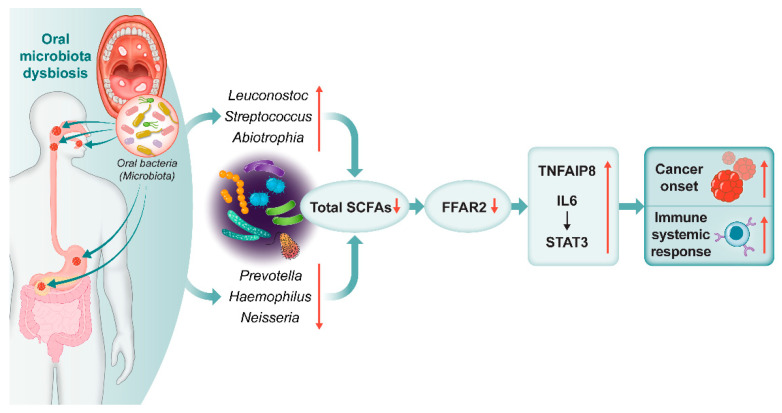
Graphical abstract. Oral microbiota-related SCFAs induce cancer and immune responses. Alcohol consumption and smoking modified the oral microbiota and short-chain fatty acids (SCFAs), free fatty acid receptor 2 (FFAR2), and relevant cytokine concentrations, which contributed to cancer initiation and systemic immune responses. **↑**: increased, **↓**: decreased.

**Table 1 cancers-15-02898-t001:** General characteristics of 745 controls and 309 cancer patients in this study.

Variable	Control	Cancer	*p* Value ^a^
	(n = 745)	(n = 309)	
Age ^a^	57.3 ± 9.20	63.2 ± 12.2	<0.0001
Sex ^a^			<0.0001
Female	434 (58.3%)	116 (37.5%)	
Male	311 (41.7%)	193 (62.5%)	
BMI (kg/m^2^)	24.0 ± 3.03	23.6 ± 3.67	<0.0001
<18.5	17 (2.28%)	24 (7.77%)	0.002
18.5~22.9	272 (36.5%)	101 (32.7%)	
23.0~24.9	197 (26.4%)	78 (25.2%)	
25.0~29.9	226 (30.3%)	90 (29.1%)	
30≥	27 (3.62%)	11 (3.56%)	
Smoking ^b^			<0.0001
Non-smoker	447 (60.0%)	152 (49.2%)	
Former smoker	202 (27.1%)	85 (27.5%)	
Current smoker	71 (9.53%)	66 (21.4%)	
Drinking ^c^			<0.0001
Non-drinker	179 (24.0%)	118 (38.2%)	
Former drinker	85 (11.4%)	56 (18.1%)	
Current drinker	441 (59.2%)	129 (41.8%)	
Stage ^d^			
1		89 (28.8%)	
2		43 (13.9%)	
3		43 (13.9%)	
4		101 (32.7%)	
T Stage ^e^			
T1		96 (31.1%)	
T2		57 (18.5%)	
T3		36 (11.7%)	
T4		63 (20.4%)	
N Stage ^f^			
N0		141 (45.6%)	
N1		40 (12.9%)	
N2		32 (10.4%)	
N3		14 (4.53%)	

Abbreviations: BMI = body mass index, T Stage = T describes the size of the tumor and any spread of cancer into nearby tissue, N Stage = N describes spread of cancer to nearby lymph nodes. ^a^
*p* values from *t*-tests for continuous variables and chi-square tests for categorical variables for comparisons between cancer and control groups. ^b^ Smoking status was split into three groups, non-smokers, former smokers, and current smokers. ^c^ Drinking status was split into three groups, non-drinkers, former drinkers, and current drinkers. ^d^ Stage information data from cancer patients. ^e^ T stage data from GC patients were not collected; only 229 cancer patients were included. ^f^ N stage information from PC and GC patients was not collected; only 170 cancer patients were included.

**Table 2 cancers-15-02898-t002:** Logistic regression analysis of six genera for all cancer risks in the GBM model.

Taxon Name	Logistic Regression	Number of Subjects ^a^		OR ^b^ (95% CI)	*p* Value ^c^
		Control	Cancer		
*Streptococcus*	Continuous scale	745	309	3.13 (2.51, 3.94)	3.33 × 10^−23^
	Quartile 1	187	33	ref	
	Quartile 2	186	36	1.09 (0.65, 1.83)	0.72
	Quartile 3	186	57	1.73 (1.08, 2.81)	0.02
	Quartile 4	186	183	5.57 (3.69, 8.62)	1.60 × 10^−15^
*Haemophilus*	Continuous scale	745	309	0.42 (0.35, 0.51)	3.30 × 10^−19^
	Quartile 1	187	165	ref	
	Quartile 2	186	46	0.28 (0.18, 0.40)	9.15 × 10^−11^
	Quartile 3	186	38	0.23 (0.15, 0.34)	1.83 × 10^−12^
	Quartile 4	186	60	0.36 (0.25, 0.52)	3.76 × 10^−8^
*Prevotella*	Continuous scale	745	309	0.51 (0.42, 0.62)	1.62 × 10^−11^
	Quartile 1	187	131	ref	
	Quartile 2	186	70	0.53 (0.37, 0.76)	0.00
	Quartile 3	186	60	0.46 (0.31, 0.66)	3.42 × 10^−5^
	Quartile 4	186	48	0.36 (0.24, 0.53)	4.55 × 10^−7^
*Leuconostoc*	Continuous scale	745	309	2.23 × 10^41^ (1.24 × 10^26^, 2.06 × 10^5^)	3.06 × 10^−6^
	=0	715	207	ref	
	>0	30	102	11.74 (7.69, 18.42)	1.45 × 10^−28^
*Neisseria*	Continuous scale	745	309	0.72 (0.63, 0.83)	4.86 × 10^−6^
	Quartile 1	187	111	ref	
	Quartile 2	186	74	0.67 (0.46, 0.95)	0.02
	Quartile 3	186	63	0.57 (0.39, 0.82)	0.00
	Quartile 4	186	61	0.55 (0.37, 0.79)	0.00
*Abiotrophia*	Continuous scale	745	309	658.08 (98.59, 5651.18)	3.24 × 10^−10^
	=0	368	104	ref	
	>0	377	205	1.92 (1.46, 2.54)	3.42 × 10^−6^

Abbreviations: OR = Odd ratio, CI = confidence interval. ^a^ The number and percentage of subjects in each category. ^b^ Odds ratio: if categorized, the first category was the standard category and was referenced for other regression coefficient calculations. ^c^
*p* value represents significance of the regression coefficient.

**Table 3 cancers-15-02898-t003:** Analysis of 3 orthologs with significant differences between cancer patients and control groups.

Function Name	Abundance Median			Fold Change	Univariate Logistic Regression	OR (95% CI) ^b^	*p* Value ^c^	AUC
	Control (n = 750)	Cancer (n = 313)	*p* Value ^a^	Cancer/ Control				
G protein-coupled receptor kinase (K08291)	3.1 × 10^−8^	1.7 × 10^−7^	1.5 × 10^−16^	5.40	Continuous Scale	1.75 (1.54–1.98)	7.2 × 10^−18^	0.73
					Quartile 1	ref	7.1 × 10^−25^	
					Quartile 2	1.37 (0.87–2.15)		
					Quartile 3	2.34 (1.57–3.49)		
					Quartile 4	3.13 (2.20–4.45)		
Futalosine hydrolase (K11783)	2.3 × 10^−8^	1.2 × 10^−7^	6.9 × 10^−8^	5.20	Continuous Scale	1.39 (1.24–1.56)	1.9 × 10^−8^	0.76
					Quartile 1	ref	6.8 × 10^−19^	
					Quartile 2	2.35 (1.58–3.51)		
					Quartile 3	3.06 (2.09–4.48)		
					Quartile 4	4.61 (3.24–6.57)		
H+-transporting ATPase (K01535)	5.1 × 10^−8^	2.5 × 10^−7^	8.5 × 10^−23^	5.00	Continuous Scale	1.74 (1.56–1.94)	1.2 × 10^−23^	0.73
					Quartile 1	ref	3.1 × 10^−25^	
					Quartile 2	1.66 (1.07–2.60)		
					Quartile 3	1.78 (1.15–2.76)		
					Quartile 4	6.74 (4.71–9.66)		

^a^ *p* value was computed using Wilcoxon rank-sum test for continuous variables. ^b^ quartiles of each genus were divided based on the distribution among controls only. ^c^
*p* value was computed using chi-square test for continuous scale and quartiles. OR: Odds ratio. 95% CI: Confidence interval.

**Table 4 cancers-15-02898-t004:** Spearman correlation of 6 genera and 3 orthologs.

Name	*Leuconostoc*		*Streptococcus*		*Abiotrophia*		*Prevotella*		*Haemophilus*		*Neisseria*	
	r ^a^	*p* ^b^	r ^a^	*p* ^b^	r ^a^	*p* ^b^	r ^a^	*p* ^b^	r ^a^	*p* ^b^	r ^a^	*p* ^b^
G protein-coupled receptor kinase (K08291)	0.16	<0.0001	0.16	<0.0001	0.05	0.13	−0.07	0.02	−0.10	0.001	−0.12	0.000
Futalosine hydrolase (K11783)	0.13	<0.0001	0.16	<0.0001	0.12	0.000	−0.14	<0.0001	−0.10	0.001	−0.01	0.79
H+−transporting ATPase (K01535)	0.22	<0.0001	0.21	<0.0001	0.07	0.02	−0.12	<0.0001	−0.18	<0.0001	−0.14	<0.0001

^a^ r value represents the relationship between 6 genera and 3 orthology. ^b^
*p* value indicates the significance of the r value.

## Data Availability

The data that support the findings of this study are available from the corresponding author upon reasonable request.

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
