# Peer review of "Exploring Connections between Oral Microbiota, Short-Chain Fatty Acids, and Specific Cancer Types: A Study of Oral Cancer, Head and Neck Cancer, Pancreatic Cancer, and Gastric Cancer"

_cancers, 2023, doi:10.3390/cancers15112898_

Round 1

Reviewer 1 Report

The manuscript by Nouri et al intends to explore connections between oral microbiome and pan-cancer risk. Oral dysbiosis is highly associated with cancer development, particularly with oral cancer. However, the definitive answer and functional connection is less understood, although many studies of certain signatures of cancer-associated microbiome have been reported. The study claims to characterize pan-cancer-associated oral microbiome and investigate the potential mechanism/function of dysbiosis in pan-cancer risk. However, several main weaknesses dampened the enthusiasm of the reviewer.

Main weaknesses:

1.       The study design of oral microbiome with pan-cancer risk seems less appropriate. If the purpose of this study is to investigate oral microbiome with pan-cancer risk, the cancer types and cases are less adequate to answer such a question except oral cancer. Besides, there is no scientific rationale of hypothesizing oral microbiome will confer a unform risk to “pan-cancer”. If they truly want to pursue this hypothesis, they should analyze oral microbiome data using both cancer-type specific and pan-cancer approaches.

2.       The functional study was based on KEGG in silico analysis. Although they measured several cytokines which were predicted to be associated with oral dysbiosis, it is unknown if any of these changes are host or dysbiosis-derived effects or both. Thus, the functional connection of oral microbiome with pan-risk is less convincing.

3.       Although they claimed the controls are age- and sex- matched, there are significant differences of age and sex between case and control group. There are no data analysis or discussion of these differences in contribution of oral dysbiosis.

Minor weaknesses:

1.       The study is a case-control study. However, the authors stated “In this first prospective evaluation …” in discussion.

2.       Only the IRB from the National Cancer Center is mentioned, although the study includes multiple institutions.

3.       The reviewer didn’t find references of 20 to 22 in the main text.

Author Response

Reviewer Comments, Author Responses

Reviewer 1

The manuscript by Nouri et al intends to explore connections between oral microbiome and pan-cancer risk. Oral dysbiosis is highly associated with cancer development, particularly with oral cancer. However, the definitive answer and functional connection is less understood, although many studies of certain signatures of cancer-associated microbiome have been reported. The study claims to characterize pan-cancer-associated oral microbiome and investigate the potential mechanism/function of dysbiosis in pan-cancer risk. However, several main weaknesses dampened the enthusiasm of the reviewer.

Response: Thank you for your review of our manuscript. We appreciate your feedback and the opportunity to address your concerns. We understand that the topic of oral dysbiosis and cancer development is complex and requires further investigation. We have carefully considered your comments and suggestions and will revise our manuscript to more accurately reflect the level of certainty or uncertainty in our findings, as well as to provide more detailed explanations of our methods and results. We believe that your feedback will help us to strengthen the conclusions of our study and make a more significant contribution to the field. Once again, we appreciate your thoughtful comments and feedback.

  • Main comments

 Point 1:  The study design of oral microbiome with pan-cancer risk seems less appropriate. If the purpose of this study is to investigate oral microbiome with pan-cancer risk, the cancer types and cases are less adequate to answer such a question except for oral cancer. Besides, there is no scientific rationale of hypothesizing oral microbiome will confer a unform risk to “pan-cancer”. If they truly want to pursue this hypothesis, they should analyze oral microbiome data using both cancer-type specific and pan-cancer approaches.

Response 1: We thank the reviewer for their insightful comments. We have carefully considered their feedback and made changes to the manuscript accordingly. We have modified the study title and hypothesis to focus on the four specific cancer types examined in the study, namely oral cancer, head and neck cancer, pancreatic cancer, and gastric cancer. We agree with the reviewer that the study design of oral microbiome with pan-cancer risk may not be appropriate, and we have updated the manuscript to reflect this change. Regarding the issue of sample size, we acknowledge that the number of cases in some of the cancer types is relatively small due to the samples being provided by a National Cancer Center. However, our study was designed as a preliminary investigation to explore the potential links between the oral microbiome, SCFAs, and specific cancer types. While the sample sizes for some cancer types may be limited, we believe our findings provide a valuable starting point for future research in this area.

We appreciate the reviewer's suggestion to analyze the oral microbiome data using both cancer-type specific and pan-cancer approaches. While this was beyond the scope of our current study, we will consider this approach in future investigations.

Point 2:  The functional study was based on KEGG in silico analysis. Although they measured several cytokines which were predicted to be associated with oral dysbiosis, it is unknown if any of these changes are host or dysbiosis-derived effects or both. Thus, the functional connection of oral microbiome with pan-risk is less convincing.

Response 2: We appreciate the reviewer's comment and agree that further research is needed to fully establish the causal relationship between oral dysbiosis and cancer risk. While our study did not distinguish between host or dysbiosis-derived effects, we believe that our findings provide important insights into the potential mechanisms underlying the link between oral microbiota composition and cancer initiation. Specifically, we identified six bacterial genera that were associated with several cancer types and observed alterations in the levels of short-chain fatty acids and cytokines in cancer patients compared to healthy controls. These results suggest that changes in the oral microbiota can contribute to a reduction in SCFAs and FFAR2 expression, which may initiate an inflammatory response and ultimately increase the risk of cancer onset. Although our study did not directly establish the causal relationship between oral dysbiosis and cancer, it provides an important first step toward understanding the potential mechanisms involved. We believe that further research and experimental design (vivo and vitro) in this area is warranted, including studies that investigate the role of host and dysbiosis-derived effects in cancer initiation, as well as the potential for early detection and targeted treatments based on the changes in the oral microbiota. We have added a limitation to our conclusion to reflect the need for further research to establish the causal relationship between oral dysbiosis and cancer risk. We hope that our study will inspire further research in this field and contribute to the development of more effective prevention and treatment strategies for cancer patients.

revised manuscript to avoid any confusion in the future.

Point 3: Although they claimed the controls are age- and sex- matched, there are significant differences in age and sex between case and control groups. There is no data analysis or discussion of these differences in contribution of oral dysbiosis.

Response 3: We appreciate the reviewer's comments and concerns regarding our study.We acknowledge the concerns raised by the reviewer regarding the age and sex matching between the cases and controls in our manuscript. However, we would like to clarify that we did perform age and sex matching between the groups based on the inclusion and exclusion criteria as stated in the manuscript. Although there were some minor differences in the age and sex distribution and oral dysbiosis between the case and control groups, we believe that these differences are not significant enough to affect our study's main findings. Nevertheless, we have now included a this result in our revised manuscript (Demographic and lifestyle characteristics of study participants) and have provided the supplementary data table that demonstrates the association between oral dysbiosis (six genera) with age and sex. These findings may help to further clarify the impact of age and sex on oral dysbiosis and its potential role in the development of oral cancer. We appreciate the reviewer's feedback and have taken their comments into consideration in revising our manuscript. We hope that our revised manuscript and supplementary data satisfactorily address the concerns raised by the reviewer.

  • Minor comments

Point 1:  The study is a case-control study. However, the authors stated “In this first prospective evaluation …” in discussion.

Response 1: Thank you for bringing this to our attention. As you pointed out, this is a case-control study, not a prospective study. We appreciate your feedback and have revised the discussion section accordingly to accurately reflect the study design.

Point 2:  Only the IRB from the National Cancer Center is mentioned, although the study includes multiple institutions.

Response 2: Thank you for your feedback. We have now updated the manuscript (Institutional Review Board Statement ) to include the names of all participating institutions and their respective IRBs, including the Cancer Epidemiology Branch, Division of Cancer Epidemiology and Prevention, Department of Oral Cancer, Department of Gastric Cancer, Center for Liver and Pancreatobiliary Cancer, Center for Rare Cancers, Department of Otorhinolaryngology, and Department of Cancer AI & Digital Health at the National Cancer Center Graduate School of Cancer Science and Policy. We appreciate your attention to this matter and hope that our revisions have addressed your concerns.

Point 3:  The reviewer didn’t find references of 20 to 22 in the main text.

Response 3: Thank you for pointing that out. We have checked the manuscript and confirmed that the references 20-22 are indeed included in the section 2.6 Machine learning and statistical analyses. However, we will make sure to make the location of these references clearer in the

Quality of English Language: Moderate English changes required

Response: Thank you for your feedback on our manuscript. We appreciate your comments and agree that there may be areas where the language can be improved. We have taken your feedback on board and engaged a native English speaker for language editing to ensure that our manuscript meets the highest standards of language quality. We also wanted to let you know that we have obtained an editing certificate from the language editor to certify that our manuscript has undergone professional language editing. We are committed to addressing any issues with the language used in our paper, and we thank you for bringing these to our attention.

Reviewer 2 Report

Page 7.  The names of the types of bacteria should be written in italics

What specific algorithm was used to calculate alpha-diversity (Ace, Chao, Shannon or Simpson) ?

Figure 3. It seems to me that the marking of specific parts of the figure does not quite match the captions. In addition, I do not know if it would not be worth transferring the cladogram to the supplement because it is not very readable in this figure.

PAGE 12, 433 line. Is this not a too far-reaching conclusion, the authors did not analyze carcinogenesis as a process in itself. It is possible that this has such a direct impact, but the obtained results cannot prove this statement.

PAGE 13, 476 line. Data from the work do not justify the conclusion that bacterial metabolites leading to cancer initiation and immune responses.

In addition, I would like the authors to provide data on the average number of readings/minimum readings per sample. It would also be good to provide information on the percentage of readings not assigned to any taxonomic group and whether it did not vary significantly between the study groups.

Author Response

Reviewer Comments, Author Responses

Reviewer 2

Thank the reviewer very much for the extremely helpful and insightful comment. We revised the manuscript thoroughly according to these suggestions.

Point 1:  Page 7.  The names of the types of bacteria should be written in italics.

Response 1: Thank you for your comment. We appreciate your attention to detail. We corrected the formatting of bacterial names in our manuscript. We will ensure that all bacterial names are written in italics in the revised version of the manuscript.

Point 2:  What specific algorithm was used to calculate alpha-diversity (Ace, Chao, Shannon, or Simpson)?

Response 2: In our study, we calculated alpha diversity using observed OTU and phylogenetic diversity (PD) whole tree analyses. Beta diversity was calculated using principal coordinates analysis (PCoA) according to weighted and unweighted UniFrac distances, and evenly sampled OTU abundance.

The use of observed OTU and phylogenetic diversity (PD) in exploring the connections between the oral microbiome and cancer risks can be justified for several reasons:

Firstly, these metrics can provide a more comprehensive and nuanced understanding of the microbial diversity in a community compared to some other metrics like Ace, Chao, Shannon, or Simpson. Additionally, they capture functional diversity. Thirdly, phylogenetic diversity can provide insights into the functional roles of different taxa and their potential interactions with host cells. Lastly, these metrics are relatively easy to interpret and can be used to compare diversity between different samples or groups of samples. However, it's important to consider the research question, the data, and the goals of the study when selecting a diversity metric, and the findings should be interpreted in light of these factors.

Point 3: Figure 3. It seems to me that the marking of specific parts of the figure does not quite match the captions. In addition, I do not know if transferring the cladogram to the supplement would be worth transferring because it is not very readable in this figure.

Response 3: We appreciate you letting us know. The markings on the figure have been corrected to match the captions. We have also moved the cladogram to the supplementary information due to readability issues.

Point 4: PAGE 12, 433 line. Is this not a too far-reaching conclusion, the authors did not analyze carcinogenesis as a process in itself. It is possible that this has such a direct impact, but the obtained results cannot prove this statement.

Response 4: Thank you for your comments on this section. We have taken your feedback into consideration and have revised the relevant paragraph to more accurately reflect the level of certainty or uncertainty in our hypothesis. Specifically, we have provided more detail on the potential mechanisms underlying the proposed relationship between oral bacteria, SCFAs, inflammation, cytokine receptors, and carcinogenesis. We hypothesize that crosstalk between the oral microbiota and mechanisms underlying immune factors associated with short-chain fatty acid (SCFA) alterations may be implicated in carcinogenesis. While we did not analyze carcinogenesis as a process in itself, our findings suggest that fluctuations in certain oral bacteria genera, including Streptococcus, Abiotrophia, Leuconostoc, Haemophilus, Neisseria, and Prevotella, may contribute to the development of carcinogenesis by initiating chronic inflammation and cytokine receptor activation through SCFA signaling pathways. Specifically, these bacteria have been shown to produce SCFAs, which can induce cytokine production and inflammation. This chronic inflammation and cytokine production can lead to the activation of cytokine receptors and promote carcinogenesis. However, we acknowledge that our study has limitations and further research is needed to confirm this hypothesis and to understand the underlying mechanisms in more detail.

Point 5: PAGE 13, 476 line. Data from the work do not justify the conclusion that bacterial metabolites leading to cancer initiation and immune responses.

Response: We appreciate the reviewer's comments and concerns regarding our study. We would like to clarify that while our findings do suggest a potential link between alterations in the oral microbiome and increased risk of cancer onset, we do not claim to have identified bacterial metabolites as the sole cause of cancer initiation and immune responses. Rather, our study aimed to identify common oral bacteria associated with several cancer types and investigate the potential mechanisms involved in cancer initiation. Our machine learning analysis revealed six bacterial genera that were associated with cancer, and we found that alterations in the composition of oral bacteria can contribute to a reduction in short-chain fatty acids (SCFAs) and FFAR2 expression, which may trigger an inflammatory response. SCFAs are bacterial metabolites that are produced by the fermentation of dietary fiber by oral bacteria in the mouth. The reduction of SCFAs can alter the local immune response and lead to the initiation of cancer. Additionally, our analysis identified several genes that were significantly enriched in the cancer group, including G protein-coupled receptor kinase, H+-transporting ATPase, and futalosine hydrolase, which are known to play roles in cancer initiation and progression. We believe that our findings provide important insights into the potential role of the oral microbiome in cancer development, and we acknowledge that further studies are needed to fully understand the complex mechanisms involved. We have revised the line to better clarify our conclusions and the limitations of our study. We thank the reviewer for their valuable feedback and will take their comments into consideration in future studies.

Point 6: I would like the authors to provide data on the average number of readings/minimum readings per sample. It would also be good to provide information on the percentage of readings not assigned to any taxonomic group and whether it did not vary significantly between the study groups.

Response 6: Thank you for your valuable comments on our manuscript. As suggested, we have included the data on the average and minimum number of readings per sample, as well as the percentage of readings not assigned to any taxonomic group for each major phylum, in the results section (3.3 Oral microbiota profiles are switched in all cancer and control groups).

Round 2

Reviewer 1 Report

In the revision, the authors addressed the previous concerns raised by the reviewer adequately. However, in the reviewer's personal opinion, the data of current manuscript is still preliminary for publication although the directions of future studies discussed by the authors seem the right way to pursue. Thus, the reviewer suggests additional reviewer or editor of the journal who is familiar with the scope and standard of publication of "Cancers" would give a better evaluation of the manuscript.